# Statistical Meta-Analysis of Risk Factors for Endometrial Cancer and Development of a Risk Prediction Model Using an Artificial Neural Network Algorithm

**DOI:** 10.3390/cancers13153689

**Published:** 2021-07-22

**Authors:** Suzanna Hutt, Denis Mihaies, Emmanouil Karteris, Agnieszka Michael, Annette M. Payne, Jayanta Chatterjee

**Affiliations:** 1Academic Department of Gynaecological Oncology, Royal Surrey NHS Foundation Trust Hospital, Guildford GU2 7XX, UK; s.hutt@surrey.ac.uk (S.H.); a.michael@surrey.ac.uk (A.M.); jayanta.chatterjee1@nhs.net (J.C.); 2Department of Clinical and Experimental Medicine, Faculty of Health and Medical Sciences, School of Biosciences and Medicine, University of Surrey, Guildford GU2 7XH, UK; 3Department of Computer Science, College of Engineering, Design and Physical Sciences, Brunel University, London UB8 3PN, UK; denismihaies@yahoo.com; 4Department of Life Sciences, Division of Biosciences, College of Health, Medicine and Life Sciences, Brunel University, London UB8 3PN, UK; emmanouil.karteris@brunel.ac.uk; 5Department of Cancer and Surgery, Imperial College London, London SW7 2BX, UK

**Keywords:** endometrial cancer, risk, neural network

## Abstract

**Simple Summary:**

A robust and comprehensive meta-analysis, for the first time, identified definitely that BMI is by far the most influential risk factor in endometrial cancer. Risk factors were previously only studied individually and or in smaller meta-analysis studies which grouped some factors together. BMI was shown to be an important risk factor with other factors less so, but no rank order was established. This work also offers, for the first time, a neural network computer model to predict the overall increase or decreased risk of cancer for individual patients, which is 98.6% accurate. This prediction can be used as a tool to determine if a patient should be considered for testing and to predict diagnosis, as well as to suggest prevention measures to patients.

**Abstract:**

Objectives: In this study we wished to determine the rank order of risk factors for endometrial cancer and calculate a pooled risk and percentage risk for each factor using a statistical meta-analysis approach. The next step was to design a neural network computer model to predict the overall increase or decreased risk of cancer for individual patients. This would help to determine whether this prediction could be used as a tool to decide if a patient should be considered for testing and to predict diagnosis, as well as to suggest prevention measures to patients. Design: A meta-analysis of existing data was carried out to calculate relative risk, followed by design and implementation of a risk prediction computational model based on a neural network algorithm. Setting: Meta-analysis data were collated from various settings from around the world. Primary data to test the model were collected from a hospital clinic setting. Participants: Data from 40 patients notes currently suspected of having endometrial cancer and undergoing investigations and treatment were collected to test the software with their cancer diagnosis not revealed to the software developers. Main outcome measures: The forest plots allowed an overall relative risk and percentage risk to be calculated from all the risk data gathered from the studies. A neural network computational model to determine percentage risk for individual patients was developed, implemented, and evaluated. Results: The results show that the greatest percentage increased risk was due to BMI being above 25, with the risk increasing as BMI increases. A BMI of 25 or over gave an increased risk of 2.01%, a BMI of 30 or over gave an increase of 5.24%, and a BMI of 40 or over led to an increase of 6.9%. PCOS was the second highest increased risk at 4.2%. Diabetes, which is incidentally also linked to an increased BMI, gave a significant increased risk along with null parity and noncontinuous HRT of 1.54%, 1.2%, and 0.56% respectively. Decreased risk due to contraception was greatest with IUD (intrauterine device) and IUPD (intrauterine progesterone device) at −1.34% compared to −0.9% with oral. Continuous HRT at −0.75% and parity at −0.9% also decreased the risk. Using open-source patient data to test our computational model to determine risk, our results showed that the model is 98.6% accurate with an algorithm sensitivity 75% on average. Conclusions: In this study, we successfully determined the rank order of risk factors for endometrial cancer and calculated a pooled risk and risk percentage for each factor using a statistical meta-analysis approach. Then, using a computer neural network model system, we were able to model the overall increase or decreased risk of cancer and predict the cancer diagnosis for particular patients to an accuracy of over 98%. The neural network model developed in this study was shown to be a potentially useful tool in determining the percentage risk and predicting the possibility of a given patient developing endometrial cancer. As such, it could be a useful tool for clinicians to use in conjunction with other biomarkers in determining which patients warrant further preventative interventions to avert progressing to endometrial cancer. This result would allow for a reduction in the number of unnecessary invasive tests on patients. The model may also be used to suggest interventions to decrease the risk for a particular patient. The sensitivity of the model limits it at this stage due to the small percentage of positive cases in the datasets; however, since this model utilizes a neural network machine learning algorithm, it can be further improved by providing the system with more and larger datasets to allow further refinement of the neural network.

## 1. Introduction

Endometrial cancer is the fourth most common cancer among postmenopausal women in the United Kingdom with more than 9377 cases between 2015 and 2017 [1]. It is the most common gynecological cancer in developed countries, and it is commonly considered a “curable cancer” as approximately 75% of cases are diagnosed before the disease has spread outside the uterus [2]. It is known to be hormone-related, and many of the risk factors are linked to excessive levels of estrogen. With that said, statistics show that 34% of cases are preventable, as a large percentage of the population is unaware of the factors which raise the risk of developing this type of cancer.

The aim of this study was to determine the level of risk of six of the most commonly identified risk factors using a meta-analysis approach and then use these data to implement an algorithm within a risk prediction neural network model which uses machine learning to calculate an individual’s risk. We wish to assist clinicians to better identify patients at risk of developing endometrial cancer and help patients make an informed choice about possible actions they can take to lessen their risk. Individualized options for decreasing a patient’s risk on the basis of their risk factor data can be suggested to the patient in order to reduce the risk as much as possible. There are multiple expected benefits of a good prediction model, including the discovery of patients who are at a higher risk than normal and may benefit from targeted prevention treatment, the increasing awareness in patients about the risk factors which put them at a high risk, and the help provided to the clinicians in decision making.

Endometrial cancer statistics show that it is primarily a disease of postmenopausal women, with about 25% of cases occurring in premenopausal women, and 5% occurring in women younger than 40 years of age [3]. This type of cancer is known to be hormone-related and driven by estrogen; thus, the levels of estrogen and progesterone in a woman’s body can affect her risk of endometrial cancer. Many of the risk factors are directly or indirectly linked to a state of excessive estrogen. During menopause, the levels of estrogen and progesterone shift, with a decrease in progesterone production. When estrogen hormone is present with subnormal levels of progesterone, it can cause the endometrium to become thickened, potentially leading to endometrial cancer [3]. Consequently, protective factors seem to be related to conditions that may result in decreased estrogen exposure. As the most common tumor of the female reproductive tract, endometrial cancer remains the fourth most common cancer in women in the developed world [1], with the incidence of endometrial cancer increasing rapidly concurrently with the increasing prevalence of obesity.

Breast, endometrial, and ovarian cancers share some hormonal and epidemiologic risk factors. There are several validated models which predict absolute risk of breast cancer such as the Gail model; however, there are fewer models for ovarian cancer and endometrial cancer. One study proposed an actual risk prediction model for endometrial cancer, and, although the study presented some interesting points, the proposed model had several limitations [4]. One limitation is that the risk factors considered are given weights based on assumptions, rather than using pooled risk ratios of multiple studies which examine the relationship of the specific risk factors and endometrial cancer or the use of logistic regression. Additionally, the prevention techniques described are generic and are based on the risk category of the patient and not on the particular risk factors. A validated risk prediction model which is based on easily obtainable epidemiologic and clinical data which can accurately predict a person’s risk is, therefore, urgently required. It can assist in identifying individuals at particularly high risk of developing endometrial cancer and who may benefit from targeted primary prevention strategies, as well as determine whether endometrial testing is needed or not.

Across Europe, it has been estimated that 60% of endometrial cancer cases may be due to excess body weight. Worldwide, the prevalence of obesity (body mass index, BMI >30 kg/m^2^) has doubled in the last three decades; each year, 2.8 million people around the world die as a result of being overweight or obese. Among women, obesity is more strongly associated with the development of endometrial cancer than any other cancer type [5] and, similarly to Europe, approximately 57% of endometrial cancers in the United States are thought to be attributable to being overweight and obese [6]. This is not necessarily something surprising, as endometrial cancer is hormone-related and obesity is also closely tied to hormonal changes, including estrogen levels, considering that obese or overweight women have higher rates of circulating estrogen in their bloodstream. Despite the clear evidence of obesity associated with endometrial cancer, there is still limited public awareness of the relationship as healthcare providers are often reluctant to counsel patients with endometrial cancer about obesity.

Smoking is a great risk factor for any type of cancer, although studies show that smoking does not obviously correlate with endometrial cancer, with some studies even suggest that smoking is a protective risk factor [7]; this risk factor was, therefore, not included in this risk model.

Reproductive factors are also strongly tied to hormonal changes and were considered; this includes parity and the use of contraceptive methods. It has been demonstrated that a woman who has given at least one birth during her lifetime has a lower risk of developing endometrial cancer, as the hormonal balance shifts toward more progesterone during pregnancy [8]. On the other hand, women who have never been pregnant have a higher risk, especially if they are also infertile. For the same reason, use of the combined oral contraceptive pill (COCP) or intrauterine progesterone device (IUPD/IUS) is associated with a significant reduction in endometrial cancer risk due to suppression of endogenous estrogen levels and increased exposure to progesterone throughout the menstrual cycle [9]. Some studies have shown that polycystic ovary syndrome, which affects 6–8% of women of reproductive age, and insulin insensitivity (or resistance), which are both components of metabolic syndrome, may play a role in the pathogenesis of endometrial cancer, perhaps through hormonal disruption, which causes higher androgen and estrogen levels and lower progesterone levels [10].

Hyperinsulinemia and the insulin-resistant state are known to be associated closely with obesity. The positive association of endometrial cancer with hyperinsulinemia and type 2 diabetes is well documented, and several studies have shown an association of type 2 diabetes with endometrial cancer risk.

Hormone replacement therapy increases the levels of circulating estrogen, which is required to reduce vasomotor symptoms such as hot flushes and night sweats. This therapy may reduce postmenopausal symptoms and prevent long-term problems due to estrogen deficiency. Sequential HRT is known to increase the risk of endometrial cancer, with risk being inversely proportional to the number of days progestin is given for. Continuous combination HRT, on the other hand, has been shown not to increase endometrial cancer risk and may even reduce it, presumably because of the protective effects of progesterone on the endometrium [11]. However, endometrial cancer is a hormone-sensitive carcinoma, and the use of HRT may stimulate estrogen receptors in residual carcinoma cells. These risk factors have been strongly associated with endometrial cancer, and clear evidence of the relationship between them and endometrial cancer has been shown.

### Preventive Measures According to Cancer Research UK

Statistics from 2015 show that 34% of all cases of endometrial cancer are preventable [12]; therefore, the right prevention techniques are needed to help patients mitigate the risk of developing this type of cancer and eventually reduce this number where possible. Obesity is the most avoidable risk factor and is the leading risk factor for this and many cancer types [12]. Calorie-controlled diets, regular physical exercise in life, and adopting a Mediterranean diet [13] have been shown to help reduce the risk of endometrial cancer by losing excess weight. For morbidly obese patients, bariatric surgery may be the best option as it produces significant and durable results [14]. These lifestyle modifications also help to ameliorate polycystic ovary syndrome, which is another risk factor of endometrial cancer. Medications such as metformin which have been shown to help prevent and treat type 2 diabetes and PCOS also lower the risk in women who are prone to endometrial cancer [15,16]. However, as diabetes and PCOS are strongly associated with obesity, it is important that the patient reduces their BMI significantly to lower their overall risk, in addition to taking medications.

Given the growing importance of predictive medicine, there is a growing reliance on machine learning to make diagnostic predictions. Most studies using machine learning in the medical arena targeted breast cancer and were concerned with three clinical endpoints: (1) the prediction of cancer susceptibility, (2) the prediction of cancer recurrence, and (3) the prediction of cancer survivability [17]. Recently, machine learning algorithms, which used a set of clinical features including risk factors as input, were used to make predictions on breast cancer patients [18]. The algorithms showing the best accuracy in these studies were the artificial neural network and the K-nearest neighbor algorithms, with almost all reported studies concerning cancer prediction using an artificial neural network as their primary predictor, as it has the ability to learn and model nonlinear and complex relationships [17]. Furthermore, these studies showed that the use of artificial neural networks could substantially improve the accuracy of cancer susceptibility and cancer outcome prediction relative to simple statistical methods.

In this study, we wished to determine the rank order of risk factors for endometrial cancer and calculate a pooled risk and percentage risk for each factor using a statistical meta-analysis approach. Then, a neural network computer model was designed to predict the overall increase or decreased risk of cancer for particular patients. This prediction can be used as a tool to determine if a patient should be considered for screening testing and to predict diagnosis, as well as to suggest prevention measures to patients.

## 2. Methods

### 2.1. Meta-Analysis and Determination of Pooled Risk

A meta-analysis of data from articles that studied the relationship of endometrial cancer and six chosen risk factors (obesity, contraception use, hormone-replacement therapy (HRT), type 2 diabetes, polycystic ovary syndrome (PCOS), and parity) was conducted. The total number of articles identified from a literature search using these key words prescreening was 9463. Articles were then screened and selected according to strict criteria, and the data therein were categorized as follows:
**Inclusion Criteria****Exclusion Criteria**Endometrial cancerRisk factorsWomenPublication dates 2003–2019English languageOther cancersLynch syndromeOther genetics/gene mutationsMortality risksNonhuman<2003Non-EnglishReviews without dataCase reportsLetters, news, notes, commentaries, and editorialsConference abstractsBook chaptersFamily history

On further reviewing papers, those that reported on particular risk factors considered identifiable/modifiable within primary care without need for investigation (listed below) were chosen for a second screen.

Risk factors within primary care included the following:Obesity—BMI/anthropometry,Diabetes—including IGT (impaired glucose tolerance),Parity,PCOS,HRT,Contraceptives.

A total of 111 articles were then put through a second screen to be included in the meta-analysis. Articles were excluded in this second screen on the basis of the following:Not enough published data to perform analysis,No/poor controls in study design,Numbers too small (<15),Old datasets—not relevant to current clinical management,Heterogeneous data,Combined risk factor data only (unable to analyze risk factors individually).

A total of 51 articles were included in the meta-analysis with data obtained from non-randomized controlled studies (e.g., case–control, case–cohort), investigating the relationship of one or more of the risk factors above and endometrial cancer, where other factors were controlled for. Data were expressed as risk estimates such as odds ratio (OR), risk ratio (RR), hazard ratio (HR), and standard incidence ratio (SIR) with 95% confidence intervals (CIs).

Of the 51 studies that met the inclusion criteria, nine studies were related to contraceptive use (four oral and five IUD/IUPD), six studies were related to HRT (six continuous use and six noncontinuous), seven studies were related to parity (five for nulliparity and two for parity), 14 studies related to type 2 diabetes, two studies were related to polycystic syndrome (PCOS), and 18 studies were related to obesity (six for a BMI over 25, 16 for BMI over 30, and two for BMI over 40). Some studies collected data on multiple risk factors and, thus, were used in more than one calculation.

Regarding obesity, only articles that measured the body mass index (BMI) rather than other measurements of body weight were used since this is a well-established and objective measurement. According to the World Health Organization, a person is considered overweight when their BMI is over 25, obese when their BMI is over 30, and morbidly obese when is over 40 [19]. These three categories were used to subdivide data from the articles as each of them presents a different relative risk, the risk increasing with an increase in BMI.

Contraception use was divided into two categories, the use of either oral contraceptives or intrauterine progesterone device (IUPD), being the most widely used methods of contraception throughout the world, which have been shown to have a relationship with endometrial cancer risk [14,20].

Hormone-replacement therapy was divided into two categories (continuous level or cyclical level), including those who had or are currently receiving this treatment, as each therapy type is known to have a different risk of endometrial cancer incidence [19,21].

Parity was divided into two categories (nulliparous women and women who have given birth at least once). Multiple studies have shown that parity does affect the risk of endometrial cancer, and a slight decrease of risk was shown in women who have a parity of over two in comparison to women who gave birth just once [22,23]; however, this difference was not great enough to be taken into consideration.

The type 2 diabetes and polycystic ovary syndrome studies used included data from individuals who were divided into sufferers and non-sufferers.

The 51 studies used in the meta-analysis are given in the Table 1.

The risks of each factor were transformed into risk percentages to be implemented into the STATA software (https://www.stata.com/features/meta-analysis/, accessed on 24 April 2021), which was used to conduct all analyses. Forest plots were drawn to combine the risk results for each factor. The pooled risk ratios were then used to derive an absolute lifetime risk and to determine how much each risk factor contributes to the lifetime risk. According to Cancer Research UK and associated studies, the estimated lifetime risk for a female being diagnosed with endometrial cancer is about 2.8% [72], meaning that one in 36 women will develop this disease at some stage in their life. Based on that number and the results from the forest plots, we transformed the relative risk of each risk factor type into risk percentage. A relative risk that is higher than 1 means that the specific factor puts the patient at a higher risk, while a relative risk lower than 1 is considered to be protective factor. As an example, we take the risk factor contraception use with IUD for which the pooled relative risk is 0.52 (see IUD forest plot in Figure 7), meaning that it reduces the risk by 0.48 or 48%; therefore, a person’s lifetime risk of 2.8% is reduced by 48% to 1.35% if they have ever used an intrauterine device. On the other hand, a BMI ≥ 40 has a relative risk of 3.47, which, using the same method, is translated to an increase of 6.9%. Using this method, the relative risk for each risk factor was calculated.

### 2.2. Design and Implementation of a Computational Risk Prediction Model

The risk prediction model was based on a neural network algorithm. This is a classification algorithm that attempts to recognize underlying relationships in a dataset through a process that mimics the way the human brain operates. We wished to use the model to predict whether a patient with a specified set of characteristics had a high chance of developing endometrial cancer or not and, therefore, should be considered for testing, as well as to suggest any preventative measures that the patient could put in place to lessen their risk. For the creation of this model, we obtained the endometrial cancer patient dataset from the National Cancer Institute (NCI) (see Section 2.5).

The model produces two main outputs, the calculated percentage risk and the prediction of whether the individual has a high risk of having cancer or not based on this percentage. If the combination of these two is positive, the system suggests that the patient should proceed to more classical clinical investigations for endometrial cancer. We evaluated the two outputs of this risk prediction model individually to determine how well they perform and to identify possible improvements to the algorithm. After training and testing the model using known data, we evaluated the model on some blind data from 40 patients (see below) and recorded the outputs. Multiple statistical analyses were performed on the results to determine any relationship between the percentage risk and the prediction accuracy with the eventual clinical diagnosis, as well as the relationship of the percentage risk and any individual risk factor.

The model was implemented with a webpage interface (HTML was used to create the structure of the web page, CSS was used to describe the presentation of the web page, and JavaScript was used to provide the functionality needed). Visual Studio Code was used as the chosen source code editor. A neural net function (below) was used for the six risk factors mentioned earlier to predict the diagnosis of the patient.

### 2.3. Risk Factors RStudio Code

NN = neuralnet(diagnosis ~ agelevel + bmi + diabetes + contraception + hrt + parity, trainNN, hidden = c(4,2), threshold = 0.001, stepmax = 1 × 10^6^, linear.output = FALSE).

Two hidden layers with four neurons for the first and two neurons for the second hidden layer were determined to be optimal to avoid under- or overfitting. This was used to plot the neural network using the training set. The compute function was used to predict the outcomes using the test dataset, which computed the outputs of all neurons for specific arbitrary covariate vectors by the trained neural network.

### 2.4. Neural Network RStudio Code

temp_test ≤ subset(testNN, select = c(“agelevel”, ”bmi”, “diabetes”, “contraception”, “hrt”, “parity”)) NN.results ≤ compute(NN, temp_test) results ≤ data.frame(actual = testNN$diagnosis, prediction = NN.results$net.result) roundedresults ≤ sapply(results, round, digits = 0).

The results were rounded to the nearest integer. The predictions were compared to actual values using a confusion matrix in order to evaluate how well it performed on the test dataset.

The full code in R is available in Appendix A.

### 2.5. Datasets

An endometrial cancer open-source dataset was obtained from the National Cancer Institute (NCI), which contained risk factor data from each patient. We randomly selected 1200 records from this set that had data for the factors we were interested in, maintaining the same 95%–5% ratio of negative to positive cancer diagnosis as the whole dataset, giving 1142 instances (95%) with a negative diagnosis and 58 with a positive diagnosis (5%). These 1200 records were then randomly divided into a training dataset and a test dataset (70% records for training and 30% for testing), while still maintaining the 95%–5% ratio in each set.

Further data from 40 patient notes currently suspected of having endometrial cancer and undergoing investigations and treatment at Royal Surrey NHS Foundation Trust Hospital were collected to test the software. Their cancer diagnosis was not provided to the software developers.

The data from both sets was anonymized.

Ethical approval was obtained from both Royal Surrey NHS Foundation Trust Hospital and Brunel University to collect the data and conduct the study. Patients gave consent for their data to be used in this study but were not actively involved.

## 3. Results

The results of the meta-analysis of the data from studies of each risk factor type using STATA software are presented in forest plots (see Figure 1, Figure 2, Figure 3, Figure 4, Figure 5, Figure 6, Figure 7, Figure 8, Figure 9, Figure 10 and Figure 11). The forest plot displays the risk factor type, study labels, the summary data, graphical representation of the individual and overall effect sizes and their confidence intervals CIs, the corresponding values of the effect sizes and CIs, and the percentages of total weight for each study. In the graph, each study corresponds to a square, centered at a point estimate of the effect size with a horizontal line (whiskers) extending on either side of the square. The horizontal line depicts the CI. The area of the square is proportional to the corresponding study weight. The overall effect size corresponds to the diamond centered at the estimate of the overall effect size. The width of the diamond corresponds to the width of the overall CI. Note that the height of the diamond is irrelevant.

As can be seen, different studies of certain factors yielded data in high agreement with each other, e.g., diabetes and continuous HRT. However, for some factors such as PCOS, various studies yielded a wider variation of results. This is demonstrated by the longer whiskers for the individual studies and more elongated diamonds in the forest plots for studies with less agreement.

The overall or pooled relative risk and the additional percentage risk above normal are summarized in Table 2 by risk factor and type.

The forest plots allowed an overall relative risk and percentage risk to be calculated from all the risk data from the studies. The results show that the greatest percentage of increased risk is due to BMI being above 25, with the risk increasing as BMI increases. A BMI of 25 or over gives an increased risk of 2.01%, a BMI of 30 or over gives an increase of 5.24%, and a BMI of 40 or over gives an increase of 6.9%. PCOS was the second highest source of increased risk at 4.2%. Diabetes, which is incidentally also linked to an increased BMI, gave a significant increased risk along with nulliparity and noncontinuous HRT of 1.54%, 1.2%, and 0.56%. Decreased risk due to contraception is greatest with IUD at −1.34% compared to −0.9% with oral. Continuous HRT at −0.75% and parity at −0.9% also decrease the risk.

### 3.1. Training the Software Model to Correlate Percentage Risk with Diagnosis

To train the software the data for each factor for a given patient from the NCI database, the training set is entered into the software to model the risk. The training set established, by machine learning, the percentage risk boundary between a positive and negative diagnosis. A box plot was used to show the percentage risk output from the model, showing that those with a positive diagnosis have a significantly higher modeled risk than those with a negative diagnosis Figure 12).

Patients who eventually developed endometrial cancer had a median risk percentage of 6.9% with a maximum of 12%. On the other hand, patients who did not develop endometrial cancer had a median risk percentage of 4.5%, with one outlier at 6%, which was considered to be in the “medium” risk category. The mean percentage risk of those with a benign diagnosis was around 4%, whereas, for those patients with a positive diagnosis of endometrial cancer, the mean was 7%. The software was trained to classify patients into these two categories using the overall percentage risk when given data of all the different risk factors. Thus, we demonstrated that our “percentage risk” algorithm can distinguish patients with a cancer diagnosis from those with a negative diagnosis when using the risk values for each factor derived from meta-analysis data.

### 3.2. Evaluation of the Model’s Ability to Correctly Predict Diagnosis

In order to evaluate its performance, a confusion matrix was built, and several statistical measures were implemented in order to determine the level of confidence.
> table(actual, prediction)

prediction

01            actual03493129

The accuracy of the algorithm was determined by calculating the ratio of number of correct predictions to the total number of predictions made, i.e., Equation (1).
accuracy = number of correct predictions/total number of predictions(1)

Using the NCI data testing set, our results showed that the model is 98.6% accurate. This is by any means considered to be a very good accuracy value for any machine learning algorithm. Since only 5% of the individuals in the test dataset had a cancer diagnosis, the accuracy measurement of the algorithm does not show how well this model can correctly predict those who have endometrial cancer, as it could achieve a correct result by chance given the high number of individuals in the dataset with a negative diagnosis. We, therefore, decided to calculate the specificity, which is a measure of the true negative rate, corresponding to the proportion of negative individuals that were correctly predicted as negative (specificity = number of correct negative predictions/total number of negative predictions); the result was an average of 98.78% when a series of different randomly picked data were entered, which again is an excellent result.

To determine how well the algorithm could correctly predict a patient with a positive diagnosis of endometrial cancer, we calculated the sensitivity of the algorithm; the result was an average of 75% according to several runs of the algorithm (sensitivity = number of correct positive predictions/total number of positive predictions). The lower percentage was probably due to only 5% of the data having a positive diagnosis, which may not be enough for the model to be trained well enough to predict more precisely.

Considering that BMI was the biggest contributor to the modeled percentage risk, we wanted to determine the relationship between BMI alone and the modeled percentage risk calculated using all the factors of individual patients, to determine the strength of the relationship. We did this using data from a further 40 patients currently suspected of having endometrial cancer and undergoing investigations and treatment at Royal Surrey Hospital. There was a strong positive correlation between the modeled percentage risk and BMI. This is shown in a scatter plot of the risk percentage of the patients and their BMI by diagnosis (Figure 13). The plot also shows that a cancer diagnosis also positively correlated with BMI and risk percentage, as shown by the red points being clustered more to the upper part of the *x-* and *y*-axes, corresponding to a higher BMI/ risk percentage. From the plot, a BMI of >25 was associated with a cancer diagnosis with the exception of one patient. Furthermore, it could also be deduced that a negative diagnosis only correlated with a risk percentage of 5% or less, and most cancer diagnoses had a risk percentage of 6% or more. This further shows that our modeled risk percentage correlated well with diagnosis.

## 4. Conclusions

In this study, we successfully determined the rank order of risk factors for endometrial cancer and calculated a pooled risk and risk percentage for each factor using a statistical meta-analysis approach. Then, using a computer neural network model system, we were able to model the overall increase or decreased risk of cancer and predict the cancer diagnosis for particular patients to an accuracy of over 98%.

The analysis of the risk factors determined that by far the biggest risk factor is obesity with a marked increased risk as BMI increases. Obesity has been linked to many cancers and is currently recognized as the single biggest risk factor [73]. In the case of endometrial cancer, as with other cancers, there is growing evidence that obesity increases the levels of estrogen, which is a proven cell proliferation and cell turnover agent, particularly for cells with endothelial origin. Indeed, epidemiological studies have confirmed an increased risk of endometrial cancer in women with high estrogen levels [74]. Estrogen induces this endometrial proliferation through the local production of IGF-1 [75]. This rapid cell division increases the risk of genetic mutations in proto-oncogenes and tumor suppressor genes, in addition to increasing free-radical-mediated DNA damage and inhibiting apoptosis [75,76]. This, in the presence of deceased levels of progesterone (as seen in postmenopausal women when the ovaries no longer produce progesterone, but instead testosterone, or in women with PCOS), leads to the “unopposed estrogen theory”, where estrogen is not counter-balanced by the protective effects of progesterone.

Obesity leads to high levels of estrogen, as adipose tissue can convert androstenedione and testosterone into estrogen and estradiol using aromatase and 17β-hydroxysteroid dehydrogenase (17β-HSD) [77,78]; thus, estrogen production is enhanced in obese individuals.

If this increased production is not accompanied by progesterone, as is the case in postmenopausal women or women with PCOS, the risks of cancer are shown to be higher. This is due to progesterone counteracting the mitogenic effects of estrogen by raising levels of IGF- binding protein-1 (IGFPB-1), which binds excess IGF-1. This in turn increases expression of the estrogen sulfotransferase and 17β-HSD enzymes, which transform estradiol into more benign estrone [75]. This would explain why women with PCOS, who do not have the protective effects of progesterone during the luteal phase of the menstrual cycle, are at an increased risk of endometrial cancer. In contrast, users of progesterone-releasing IUDs have a significantly lower risk of endometrial cancer than nonusers [79]. Furthermore, the use of the combined oral contraceptive pill (COCP) for ≥5 years is associated with a significant reduction in endometrial cancer risk due to suppression of endogenous estrogen levels and increased exposure to progesterone throughout the menstrual cycle [9]. For the same reason, increasing parity is also a protective factor [22].

Obese women have higher insulin levels than their normal-weight counterparts; excess fat tissue reduces the responsiveness of the body to the effects of insulin; hence, levels increase to compensate. Endometrial cell proliferation is also stimulated by insulin. There is evidence for a direct effect on endometrial cancer cells of insulin and IGF-1; activation of the insulin receptor causes an increase in cell proliferation and inhibition of apoptosis [76,80,81] through both the MAPK and PI3K/Akt pathways. Insulin and IGF-1 also stimulate β-catenin, a signaling pathway involved in early tumor formation, and the Ras oncogene. Insulin increases the breakdown of IGFBP-3, thus increasing the levels of free IGF-1, promoting tumor formation. Interestingly, elevated serum insulin levels have been shown to be present in women with endometrial cancer, compared with those without the disease [74].

Obesity is characterized by chronic inflammation [82], with fat tissue producing inflammatory and carcinogenic (cancer promoting) proteins through the release of adipokines, cytokines, and sex hormone metabolism [83]; hence, obese women have elevated levels compared with normal-weight women [74]. Cytokines are produced by the activated adipocytes and infiltrating macrophages in response to expansion of the adipose tissue and localized hypoxia. Increasing BMI is associated with elevated levels of cytokines including IFNs, IL6, IL8, IL1 receptor antagonist (IL-1Ra), and C-reactive peptide (CRP) [76,84,85]. Chronic inflammation results in the generation of free radicals, as well as increased concentrations of COX2 and prostaglandin E2, leading to cell proliferation and DNA damage [86]. In addition, activation of the NF-κB pathway by inflammatory cytokines inhibits apoptosis, overcoming cell-cycle arrest, and leads to the transcription of genes encoding proinflammatory cytokines. This cycle of inflammation can result in tumor formation. Inflammation can also cause insulin resistance, and IL6 can stimulate aromatase activity and the conversion of androgens into estrogen within adipose tissue, which all contribute to the ideal conditions for tumor formation [85].

The neural network model developed in this study was shown to be a potentially useful tool in determining the percentage risk and predicting the possibility of a given patient developing endometrial cancer. The risk factors analyzed are not linked to specific histological types or the new molecular classification [87]. As such, this could be a useful tool for clinicians to use in conjunction with other biomarkers in determining which patients warrant further preventative interventions progressing to endometrial cancer. This result would allow for a reduction in the number of unnecessary invasive classical tests on patients. The model may also be used to suggest interventions to decrease the risk for a particular patient. The sensitivity of the model limits it at this stage due to the small percentage of positive cases in the datasets; however, since this model utilizes a neural network machine learning algorithm, it can be further improved by providing the system with more and larger datasets to allow further refinement of the neural network.

## Figures and Tables

**Figure 1 cancers-13-03689-f001:**
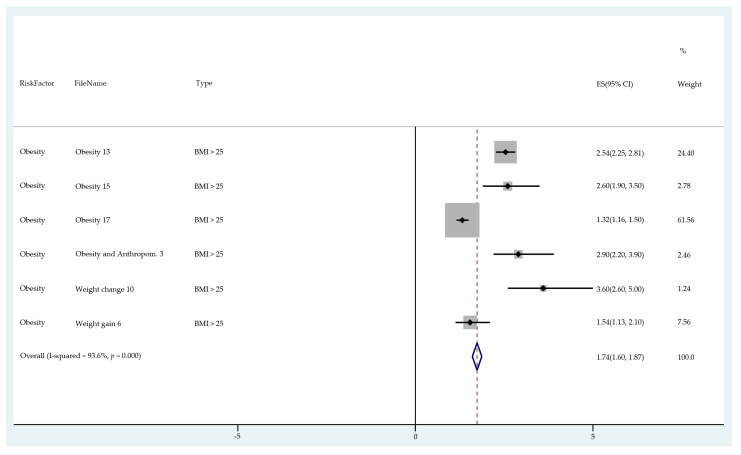
Forest plot to show risk of BMI > 25.

**Figure 2 cancers-13-03689-f002:**
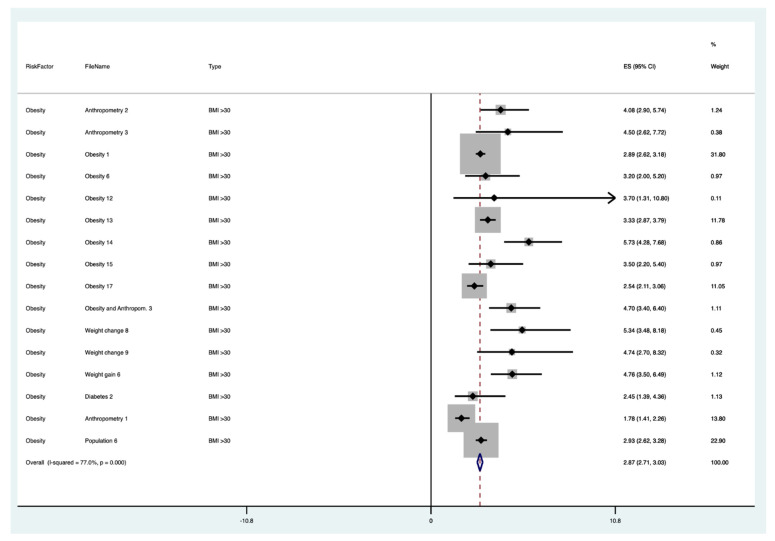
Forest plot to show risk of BMI > 30.

**Figure 3 cancers-13-03689-f003:**
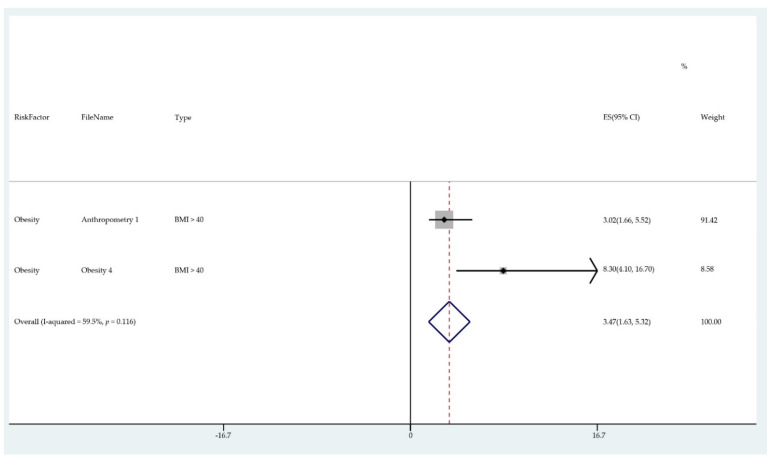
Forest plot to show risk of BMI > 40.

**Figure 4 cancers-13-03689-f004:**
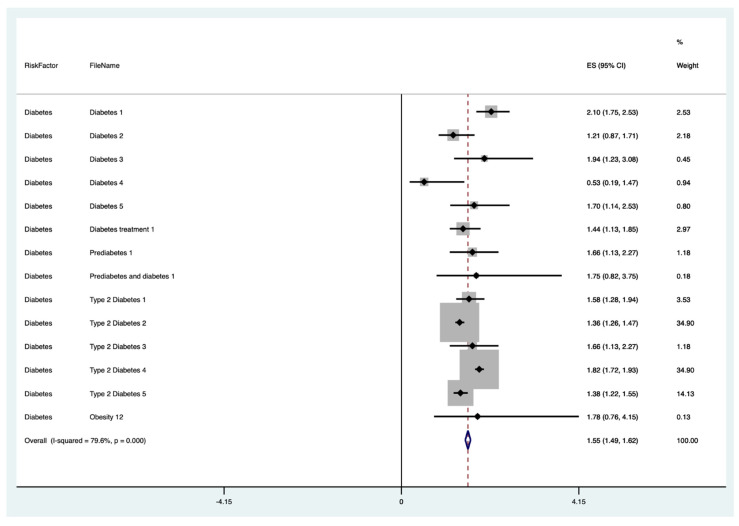
Forest plot to show risk of diabetes.

**Figure 5 cancers-13-03689-f005:**
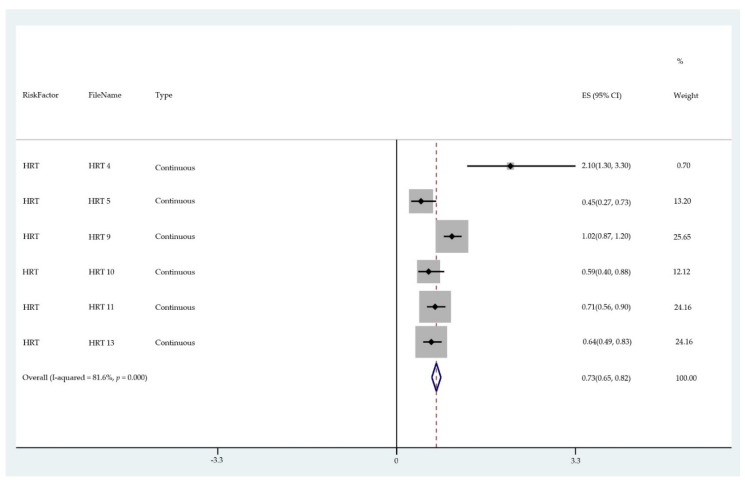
Forest plot to show risk of continuous HRT.

**Figure 6 cancers-13-03689-f006:**
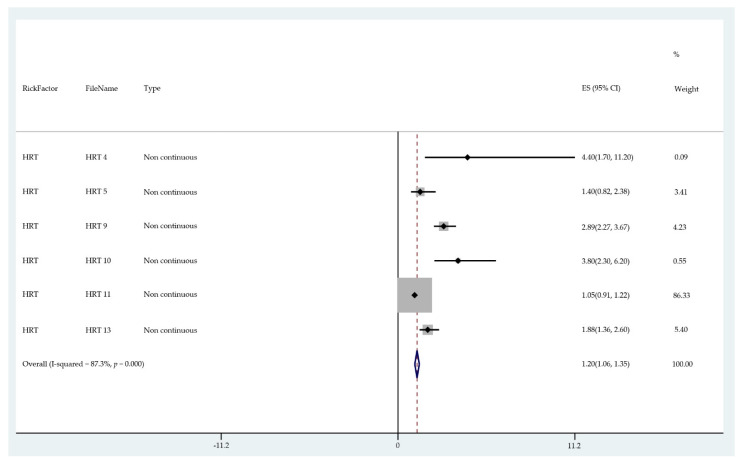
Forest plot to show risk of noncontinuous HRT.

**Figure 7 cancers-13-03689-f007:**
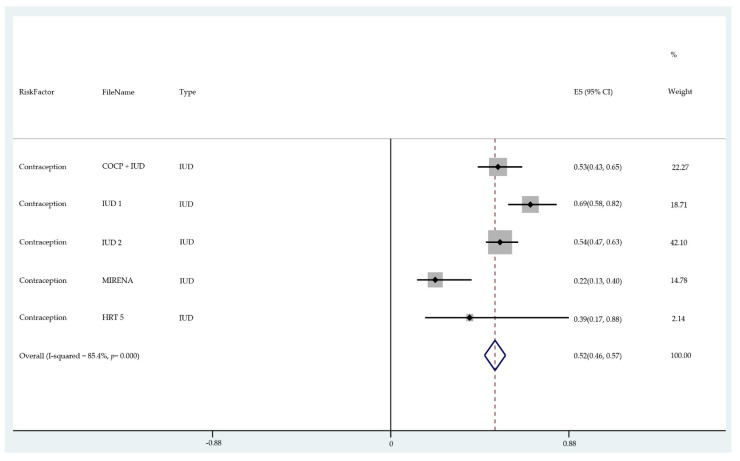
Forest plot to show risk of contraception (IUD).

**Figure 8 cancers-13-03689-f008:**
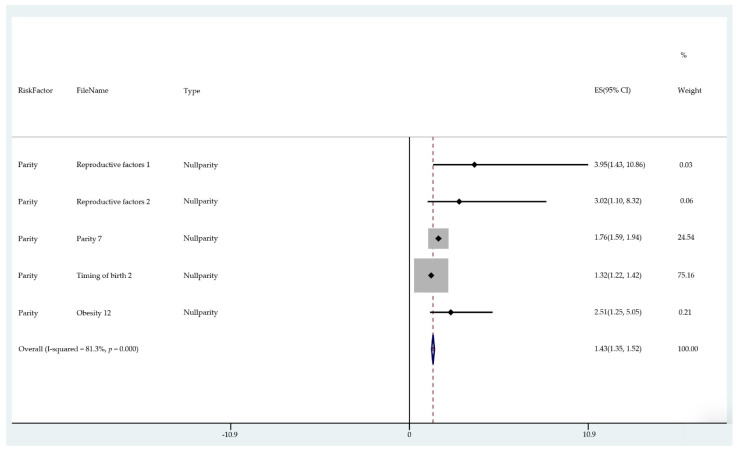
Forest plot to show risk of reproductive factors/null parity.

**Figure 9 cancers-13-03689-f009:**
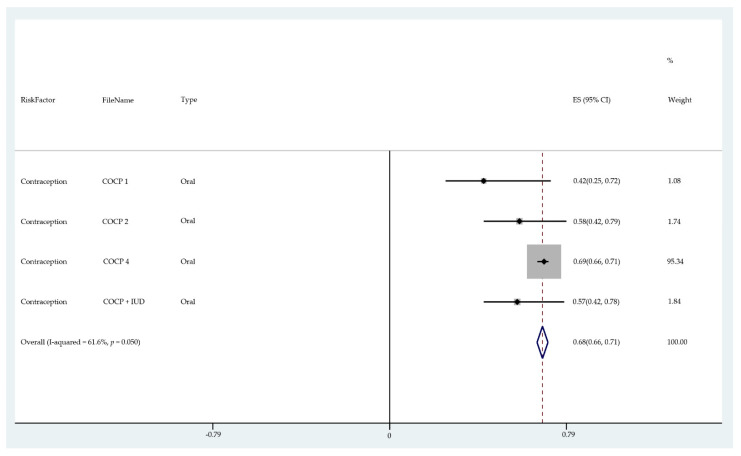
Forest plot to show risk of contraception (oral).

**Figure 10 cancers-13-03689-f010:**
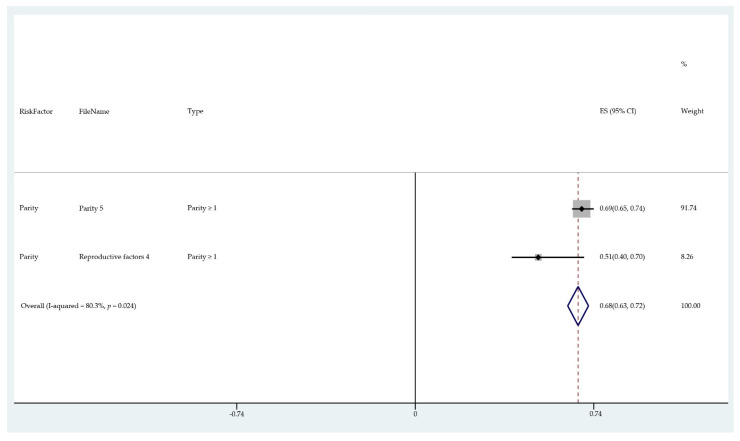
Forest plot to show risk of parity.

**Figure 11 cancers-13-03689-f011:**
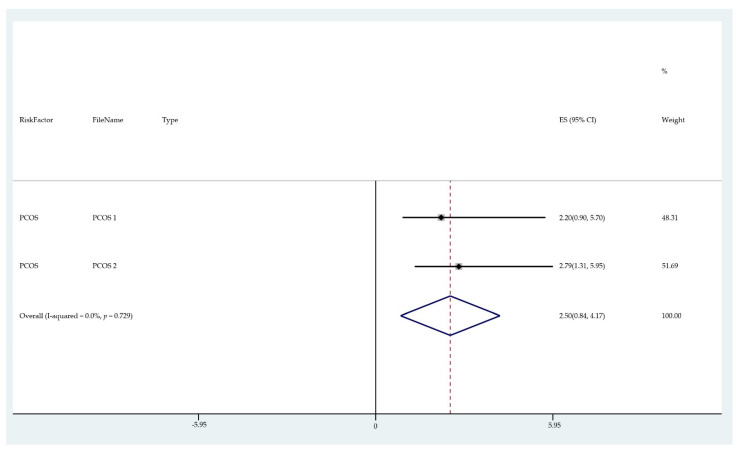
Forest plot to show risk of PCOS.

**Figure 12 cancers-13-03689-f012:**
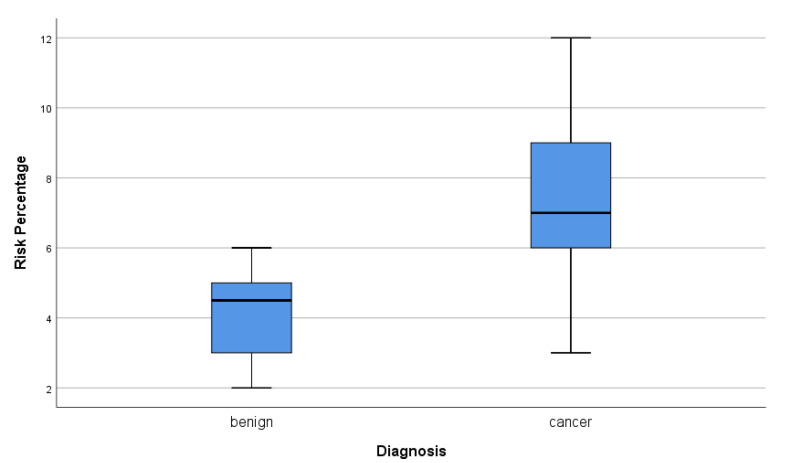
Diagnosis–risk percentage box plot.

**Figure 13 cancers-13-03689-f013:**
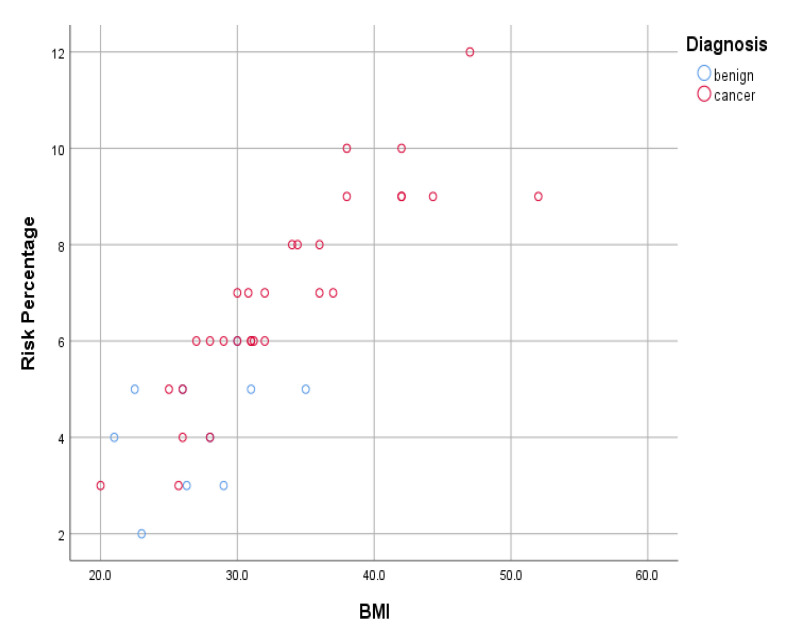
BMI–risk percentage scatter plot.

**Table 1 cancers-13-03689-t001:** Studies used in the meta-analysis of risk factors.

Study Code Number	Reference
Obesity 13	Jenabi, E., Poorolajal, J. (2015) [24]
Obesity 15	Gao, Y. et al. (2016) [25]
Obesity 17	Zhang, Y. et al. (2014) [26]
Obesity and anthropometry 3	Xu, W.H. et al. (2005) [27]
Weight change 10	Nagle, C.M. et al. (2013) [28]
Weight gain 6	Lu, L. et al. (2011) [29]
Anthropometry 2	Maso, L. et al. (2011) [30]
Anthropometry 3	Schouten LJ et al. (2004) [31]
Obesity 1	Reeves, G.K. et al. (2007) [5]
Obesity 6	Jonsson, F. et al. (2003) [32]
Obesity 12	Wise, M.R. et al. (2016) [33]
Obesity 14	Rota, M. et al. (2015) [34]
Weight change 8	Liu, Y. et al. (2016) [35]
Weight change 9	Horn-Ross, P.L. et al. (2016) [36]
Diabetes 2	Attner, B. et al. (2012) [37]
Anthropometry 1	Friedenreich, C. et al. (2007) [38]
Population 6	Yang, H.P. et al. (2012) [39]
Obesity 4	Lindemann, K. et al. (2009) [40]
Diabetes 1	Friberg, E. et al. (2007) [41]
Diabetes 3	Friberg, E. et al. (2007) [42]
Diabetes 4	Lindemann, K. et al. (2008) [43]
Diabetes 5	Bosetti, C. et al. (2012) [44]
Diabetes treatment 1	Luo, J. et al. (2014) [45]
Prediabetes 1	Huang, Y. et al. (2014) [46]
Prediabetes and diabetes 1	Lambe M et al. (2011) [47]
Type 2 diabetes 1	Johnson, J.A. et al. (2011) [48]
Type 2 diabetes 2	Lin, C.C. et al. (2014) [49]
Type 2 diabetes 3	Oberaigner, W. et al. (2014) [50]
Type 2 diabetes 4	Liu, X. et al. (2015) [51]
Type 2 diabetes 5	Lo, S.F. et al. (2013) [52]
HRT 4	Razavi, P. et al. (2010) [53]
HRT 5	Jaakkola, S. et al. (2011) [54]
HRT 9	Mørch, L.S. et al. (2016) [55]
HRT 10	Doherty, J.A. et al. (2007) [56]
HRT 11	Beral, V. et al. (2005) [19]
HRT 13	Trabert, B. et al. (2013) [57]
COCP + IUD	Tao, M.H. et al. (2006) [58]
IUD 1	Felix, A.S. et al. (2015) [59]
IUD 2	Beining, R.M. et al. (2008) [60]
MIRENA	Jareid, M. et al. (2018) [61]
Reproductive factors 1	Wernli, K.J. et al. (2006) [62]
Reproductive factors 2	Xu, W.H. et al. (2004) [63]
Parity 7	Yang, H.P. et al. (2015) [64]
Timing of birth 2	Pfeiffer, R.M. et al. (2009) [65]
COCP 1	Cook, L.S. et al. (2014) [66]
COCP 2	Hannaford, P.C. et al. (2007) [67]
COCP 4	Collaborative Group on Epidemiological Studies on Endometrial Cancer. (2015) [68]
Parity 5	Wu, Q.J. et al. (2015) [22]
Reproductive factors 4	Brinton, L.A. et al. (2007) [69]
PCOS 1	Fearnley, E.J. et al. (2010) [70]
PCOS 2	Barry, J.A. et al. (2014) [71]

**Table 2 cancers-13-03689-t002:** Pooled relative risk and percentage risk for each risk factor.

Risk Factor	Type	Pooled Relative Risk	Percentage Risk
Contraception	IUD	0.52	−1.34%
	Oral	0.68	−0.9%
HRT	Continuous	0.73	−0.75%
	Noncontinuous	1.2	0.56%
Parity	Nulliparity	1.43	1.20%
	≥1	0.68	−0.9%
PCOS	Yes	2.5	4.2%
	No	0	0%
Diabetes	Yes	1.55	1.54%
	No	0	0%
Obesity (BMI)	<25	0	0%
	≥25 and <30	1.74	2.01%
	≥30 and <40	2.87	5.24%
	>40	3.47	6.9%

## Data Availability

The data used in the study are publicly available in the references cited within the text or are available on request from the corresponding author.

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
