# Peer review of "Statistical Meta-Analysis of Risk Factors for Endometrial Cancer and Development of a Risk Prediction Model Using an Artificial Neural Network Algorithm"

_cancers, 2021, doi:10.3390/cancers13153689_

Round 1

Reviewer 1 Report

Congratulations. Your work is very interesting and innovative. Statistical methods are clear and appropiate as well as machine learning analysis.

In different parts of the manuscript the authors speak of "screening" of endometrial cancer but we know that, to date, there is no standard approved screening test for the early diagnosis endometrial cancer. Please explain better what you mean.

Line 18: “been” is repeated twice

Line 21 and 28: what do you mean by “particular patients”?

Line 27-28: please edit one “then”

Line 60-63: please modify the repetitions “model…model…model”

Line 68: you can put a more updated reference?

Line 392-397 correct, please

Introduction

Three pages of introduction are too long and distract attention. Please summarize.

Conclusion

In the conclusion there are too many notions.

Line 442- 496: Concepts already treated are repeated. Please summarize this part in order to focus more attention only on what your work communicates.

Reviewer 2 Report

The article provides meta-analysis of risk factors for cancer. The major problem with the article is overall quality of text and presentation. All figures contain completely unreadable text labels and numbers that makes it hard to even understand what is drawn there without reading the Figure caption. Article text has also a lot of technical typesetting problems. 51 references are used in the Table1 with no links to sources (but with spirituous footnote numbers that has no footnote linked to them). The description of neural network-based classification model is incomplete and do not give enough information to review it properly. So I'll suggest major revision of article to make it more readable before the article content can be evaluated from scientific point of view.

Reviewer 3 Report

This meta-analysis has identified BMI as the most influential risk factor in endometrial cancer and offers for the first time a neural network computer model to predict the overall increase or decreased risk of cancer with an accuracy of 98.6%.

The study is well conceived and the analysis is robust and well performed

Considering that the dichotomic clinical classificatiof of endometrial cancer has been replaced by a new histomolecular classification, please explain if and how this neural network computational model could be applied to determine percentage risk for individual patients, for the different molecular landscapes. Also, cite and comment this recent work: Cancers 202113(11), 2623; https://doi.org/10.3390/cancers13112623

Round 2

Reviewer 2 Report

The quality of article was substantially improved and all expressed concerns were addressed. I still suggest to increase font size on the graphs to make it more visible, but it is minor fix that can be done without another round of reviews.